# Interpretable Complex-Valued Neural Networks for Privacy Protection - ML Reproducibility Challenge 2020

## Reproducibility Summary

**Scope of Reproducibility**

The authors of the original work do not supply any code for their work. Therefore, our goal is to validate the main claims of the paper with our own implementation. The claims that we try to verify are whether the proposed complex-valued neural networks perform similar to traditional real-valued networks in classification tasks and whether the introduced method provides better protection against privacy attacks.

**Methodology**

For our own implementation we follow the author's description where possible. No explicit information about the training process and some architectures is mentioned, so we make our own assumptions where needed. Training all networks takes around 60 hours on a Nvidia RTX 2080 TI.

**Results**

In all experiments of our reproduction study, we were able to successfully validate the author's claim that the proposed network architectures provide better protection against privacy attacks. However, the observed benefits are not as extensive as suggested by the original results. We also observed strong performance degradation when using the proposed complex-valued architectures during some classification experiments. This contradicts the author's claim that the performance is on par with standard real-valued neural networks.

**What was easy**

The authors provide clear descriptions on how to transform a traditional neural network into a complex-valued neural network and how to implement the proposed complex-valued layers. Additionally, the paper does a good job at explaining how experiments are quantified.

**What was difficult**

Many implementation details are omitted in the paper, which made it difficult to get some parts to working as intended. Specifically, hyperparameter settings are not given which required us to make many assumptions. The large number of experiments also made it time consuming to test different hyperparameter settings and restricted us to only training one model per experiment.

**Communication with original authors**

Due to lack of time, we did not communicate with the authors.

# 1 Introduction

As the progress of deep learning has improved in recent years, many services utilizing deep neural networks (DNN) have been introduced to solve a large variety of problems. In order to use larger and more computationally expensive models, companies have opted to use cloud-based solutions where user data is sent remotely to be processed and its results are transmitted back to the user. This approach, however, introduces many privacy concerns such as man-in-the-middle attacks or users being uncomfortable sending their raw and possibly sensitive data to an unknown location. Even when part of the model is computed locally and only intermediate features are sent remotely, previous work has shown that attackers can accurately reconstruct the original input from just these features [2, 11, 3, 13].

Xiang et al. [17] propose a solution to protect against these attacks using complex-valued DNNs. Intermediate features are computed locally on the user's device and then converted from real-valued to complex-valued tensors rotated by a random angle. The complex-valued features are processed in the cloud and returned to the user who can rotate back the results in order to obtain real-valued features. This makes the rotation angle act as a user's private key. The authors introduce complex-valued alternatives to many commonly used layers and outline how to modify standard convolutional neural network (CNN) architectures such as ResNet [5], VGG [14], AlexNet [8] and LeNet [9] into their proposed complex-valued models. Their experiments present evidence that this apporach is highly effective against feature inversion and property inference attacks while reporting similar classification performance as real-valued networks.

In this work, we examine the reproducibility of the quantitative results reported by Xiang et al. Since no publicly available implementation currently exists, we write our own implementation from scratch in PyTorch. We first outline the details of our implementations for the complex-valued DNN and the attack models in Section 2 and then present results of our experiments in Section 3. We conclude by discussing the reproducibility of the author's central claims in Section 4. Our implementation is publicly available at `https://github.com/romech/fact-ai`.

## 1.1 Scope of Reproducibility

As the authors do not provide training details in their work, we do not aim to reproduce the exact reported numbers. Instead, we focus on validating the following main claims of the paper:

- The proposed complex-valued architectures maintain a similar classification performance as their real-valued counterparts.
- Reconstructing a model's input or inferring properties about the input from intermediate features is more difficult when using complex-valued DNNs.

# 2 Methodology

## 2.1 Complex-Valued DNN

The proposed model divides the layers of a standard DNN into three components: an encoder, processing module and decoder. The encoder and decoder are real-valued modules, identical to their real-valued counterparts, and are placed locally on the user's device. The intermediate processing module uses complex-valued layers and is located remotely. The model processes an input $I$ as follows:

1. Compute intermediate encoder features using the encoder $g$:

$$a = g(I)$$

2. Sample random noise $b$ and a random phase $\theta$ and transform $a$ into a complex-valued tensor:

$$x = \exp(i\theta)[a + ib] = (a \cdot \cos\theta - b \cdot \sin\theta) + i(b \cdot \cos\theta + a \cdot \sin\theta)$$

3. Compute complex-valued features using the processing module $\mathbf{\Phi}$:

$$h = \mathbf{\Phi}(x)$$

4. Apply the inverse rotation with $-\theta$ and drop the imaginary component to revert the complex-valued features back to real-valued ones:

$$f = \Re(\exp(-i\theta)h) = a \cdot \cos(-\theta) - b \cdot \sin(-\theta)$$

5. Compute the model's output using the decoder $d$:

$$\hat{y} = d(f)$$

This framework mitigates potential attacks by only transmitting encoded complex-valued features between the user and the cloud.

To ensure that the complex-valued features $h$ can be successfully decoded, the processing module uses layers designed to preserve the feature's phase $\theta$. This allows the decoder to successfully recover real-valued features when applying the inverse rotation $\exp(-i\theta)$. Following Trabelsi et al. [15], we represent complex numbers as two real-valued numbers corresponding to their real and imaginary parts. A complex-valued feature map of size $(N \times H \times W)$ is then represented as a real-valued tensor of size $(2 \times N \times H \times W)$.

For the complex-valued convolutions, we use the variant proposed by Trabelsi et al [15]. A convolution operation with a kernel $\mathbf{W} = \mathbf{W}_\Re + i\mathbf{W}_\Im$ on an input $\mathbf{X} = \mathbf{X}_\Re + i\mathbf{X}_\Im$ is defined as:

$$conv(\mathbf{X}) = (\mathbf{X}_\Re \otimes \mathbf{W}_\Re - \mathbf{X}_\Im \otimes \mathbf{W}_\Im) + i(\mathbf{X}_\Im \otimes \mathbf{W}_\Re + \mathbf{X}_\Re \otimes \mathbf{W}_\Im)$$

The bias term is dropped to keep the convolution operation phase invariant.

Similar to convolutions, we implement fully connected layers with weights $\mathbf{W} = \mathbf{W}_\Re + i\mathbf{W}_\Im$ as:

$$fc(\mathbf{X}) = (\mathbf{X}_\Re \mathbf{W}_\Re - \mathbf{X}_\Im \mathbf{W}_\Im) + i(\mathbf{X}_\Im \mathbf{W}_\Re + \mathbf{X}_\Re \mathbf{W}_\Im)$$

We implement the normalization, activation, pooling and dropout layers as described in the original work.

One of the requirements to ensure protection against an adversary, is that when rotating encoded complex-valued features by some random phase $\theta'$, which results in estimated features $a^* = \Re(\exp(-i\theta')[a + ib])$, $a^*$ is indistinguishable from the real features $a$. If this does not hold, an adversary would be able to test different angles and know exactly when they have guessed the correct rotation. To address this issue, an adversarial framework [4] is introduced to enforce the encoder to produce indistinguishable features. An additional $3 \times 3$ convolution is appended to the encoder $g$ and its outputs are passed to a discriminator $D$ consisting of a $4 \times 4$ convolution with a stride of 2 and a fully connected layer. The encoder and discriminator are trained using the Wasserstein GAN (WGAN) loss [1] which is defined as:

$$\min_g \max_D = \mathbb{E}_I[D(g(I)) - \mathbb{E}_{\theta, b \neq g(I)}[D(\Re[(g(I) + ib)\exp(i\theta)])]]$$

By approximating the second expectation with $k$ randomly selected phases $\theta$ and noise samples $b$, the authors claim that the model achieves a $k$-anonymity privacy which guarantees that any correctly decoded feature map cannot be distinguished from $k - 1$ other incorrectly decoded feature maps. In our experiments we use $k = 5$ and $b$ is a randomly selected element from the same batch. Both the WGAN loss and the loss of the model's target task are optimized simultaneously.

## 2.2 Classification Models

The original paper focuses on image classification models and we reproduce its experiments for the ResNet-20/32/44/56 [5] and LeNet [9] architectures. We implement the networks following their respective papers and divide the networks into an encoder, processing module and decoder according to the original author's description. Two ResNet variants are introduced, ResNet-$\alpha$ and ResNet-$\beta$, that differ in the position where the network is split between the decoder and processing module.

The performance of the complex-valued DNNs is compared against three baselines:

1. The original real-valued DNNs without modifications.

2. The original real-valued DNNs with random noise $\epsilon$ added to the encoder's outputs to produce features $a' = a + \gamma\epsilon$ where $\gamma$ is a scaling constant. Since the authors do not report the random noise's distribution, we use a normal distribution with unit covariance and mean equal to the average $a$ within the batch.

3. The original real-valued DNN with the additional $3 \times 3$ convolution layer introduced in the adversarial framework. This accounts for the possible performance change introduced by the additional layer.

## 2.3 Attack Models

Two types of attacks are carried out to assess the privacy protection capabilities of the proposed complex-valued DNNs: feature inversion attacks and property inference attacks.

Feature inversion attacks [2, 11] aim to reconstruct a network's original input from intermediate network features. We implement the attack model $dec$ as a U-Net [12] as described by the authors. For real-valued classification networks,

the output of the encoder is passed to the attack model. For complex-valued networks two scenarios are tested. In the naive approach, the attacker is given the encoded complex-valued features $x$ which are converted to real-valued tensors by concatenating the real and imaginary components. In the second approach, a regression model is trained to predict the rotation angle of $x$. The estimated angle is subsequently applied in a rotation to obtain estimated real-valued features $a^*$ which are used by $dec$ to reconstruct the input. Since the authors do not specify the angle prediction model's architecture, we use the same architecture as the previously mentioned discriminator. The angle prediction and inversion models are optimized with the $\ell_1$ loss and both are trained separately.

Property inference attacks [3, 13] aim to learn hidden properties about the input data from intermediate features. We implement three inference attacks used by Xiang et al.:

- **Inference attack 1**: An attack model is trained on raw images to predict hidden properties and evaluates on the reconstructions from the inversion model $dec(a^*)$.
- **Inference attack 2**: Using the angle prediction network, an attack model is trained on estimated real-valued features $a^*$ to predict hidden properties.
- **Inference attack 3**: Using the inversion model $dec(a^*)$, an attack model is trained on input reconstruction to predict hidden properties.

The attack models used to predict the hidden properties is a real-valued ResNet-56.

## 2.4 Datasets

In our experiments we use the CIFAR-10 and CIFAR-100 datasets [7]. CIFAR-10 is made up of ten classes each having 5000 training and 1000 test images of size $32 \times 32 \times 3$. During training, we apply the standard augmentations of normalization, random horizontal flip and random cropping with padding of 4. At test time we only apply normalization. CIFAR-100 labels the same images as CIFAR-10 into 100 classes each having 500 training and 100 test images. For the property inference attacks we also use the dataset's coarse labels which classifies images into 20 superclasses. We use the same data augmentation steps as with CIFAR-10.

## 2.5 Hyperparameters

ResNet models on CIFAR-10 are trained using SGD with a momentum of 0.9, an initial learning rate of 0.1 and a batch size of 128 following He et al. [5]. ResNet-56 variants with additional layers have an initial learning rate of 0.05 to stabilize training. Due to the large number of experiments, all other classification experiments are trained using Adam [6] with $\beta_1 = 0.9$, $\beta_2 = 0.999$, an initial learning rate of $10^{-3}$ and a batch size of 128. We train all classification networks for 200 epochs and decay the learning rate by a factor of 0.1 after 100 and 150 epochs. Complex networks use $k = 5$ for adversarial training.

We train the feature inversion U-Net models using Adam with $\beta_1 = 0.9$, $\beta_2 = 0.999$, a learning rate of $10^{-4}$ and a batch size of 128 for 15 epochs. The angle prediction model is trained using the same settings for 50 epochs. For the property inference attacks, both the prototype and attack classification models are trained using Adam with $\beta_1 = 0.9$, $\beta_2 = 0.999$, an initial learning rate of $10^{-3}$ and a batch size of 128. The models are trained for 200 following the same learning rate decay schedule as above.

## 2.6 Computational Requirements

We run all of the experiments on an Nvidia RTX 2080 TI with 11 GB of memory. Due to the high number of experiments we only train each model once. The baseline ResNet and LeNet models take 30 minutes to train, while the complex-valued networks take between 90 minutes and four hours depending on the network depth. The inversion attacks require less training and take around 15 minutes to train. Inference attacks take between 30 and 60 minutes to train.

# 3 Results

## 3.1 Classification Performance

We reproduce the classification experiments from the original work to investigate the author's claim that complex-valued DNNs achieve similar performance to their real-valued counterparts. Table 1 shows the classification error rates for several complex-valued ResNet variants, their baseline real-valued counterparts and the baseline DNNs with an

| | | Classification Error Rates (%) | | |
|---|---|---|---|---|
| | Dataset | Original DNN | DNN with additional layer | Complex-Valued DNN |
| ResNet-20-$\alpha$ | CIFAR-10 | 7.77 | 9.06 | 11.88 |
| ResNet-20-$\beta$ | CIFAR-10 | 8.03 | 8.30 | 13.13 |
| ResNet-32-$\alpha$ | CIFAR-10 | 7.27 | 8.95 | 11.27 |
| ResNet-32-$\beta$ | CIFAR-10 | 7.22 | 8.88 | 10.56 |
| ResNet-44-$\alpha$ | CIFAR-10 | 6.57 | 9.32 | 10.02 |
| ResNet-44-$\beta$ | CIFAR-10 | 7.31 | 8.84 | 10.14 |
| ResNet-56-$\alpha$ | CIFAR-10 | 7.19 | 8.45 | 10.03 |
| ResNet-56-$\beta$ | CIFAR-10 | 6.89 | 8.27 | 9.78 |

Table 1: Classification experiments on CIFAR-10 using several ResNet variants. We compare the complex-valued DNNs against their original counterparts and DNNs with an additional convolutional layer.

additional layer on the CIFAR-10 test set. Complex-valued DNNs perform 3 - 5 % worse than the original network. The performance degradation reported in the original work never exceeds 2% for any ResNet variant with some complex-valued networks outperforming their real-valued counterparts.

We furthermore reproduce the classification experiments with the LeNet and ResNet-56-$\alpha$ on the CIFAR-10 and CIFAR-100 datasets and compare the performance of the real-valued, noisy and complex-valued DNNs. Results can be seen in Table 2. We observe that the complex-valued LeNet performs significantly worse than the original network on both datasets, while the ResNet performs similar to what we observed on CIFAR-10. These findings contradict the reported results in the original work where the complex-valued LeNet reduced the error by roughly 2 % and the complex-valued ResNet-56-$\alpha$ on CIFAR-100 improved $\sim$ 11 % compared to the real-valued baseline. We found that while the noisy baselines perform consistently worse than the original DNNs, they all but in one experiment achieve a lower error than the complex-valued networks. The opposite was reported in the original work, with the complex-valued networks outperforming all noisy baselines by up to 30% in some experiments.

| | | Classification Error Rates (%) | | | | | |
|---|---|---|---|---|---|---|---|
| | Dataset | Original DNN | DNN with additional layers | Noisy DNN $\gamma = 0.2$ | Noisy DNN $\gamma = 0.5$ | Noisy DNN $\gamma = 1.0$ | Complex-Valued DNN |
| LeNet | CIFAR-10 | 26.09 | 26.38 | 26.73 | 29.17 | 34.98 | 33.84 |
| LeNet | CIFAR-100 | 60.89 | 60.34 | 61.55 | 64.69 | 66.83 | 74.32 |
| ResNet-56-$\alpha$ | CIFAR-100 | 32.20 | 32.46 | 32.67 | 33.58 | 33.91 | 35.30 |

Table 2: Reproduction of classification experiments on CIFAR-10 and CIFAR-100 using LeNet and ResNet-56-$\alpha$. We compare the complex-valued DNNs against their original counterparts, DNNs with an additional convolutional layer and the noisy baselines.

## 3.2 Protection Against Inversion Attacks

We measure the performance of the angle prediction model by the mean absolute error (MAE) $|\theta' - \theta|$ where $\theta'$ is the model's prediction. Results on CIFAR-10 and CIFAR-100 can be seen in Table 3. The attack model performs well, achieving $< 0.1$ radians on ResNet variants while performing worst on LeNet. This contradicts the original work where all models achieve an MAE of $\sim 0.8$ indicating that in our experiments, the complex-valued features $x$ contain discriminating information about its phase.

We measure the reconstruction performance of the feature inversion attack model by the pixel MAE. Table 4 shows results of reconstructing the input using intermediate network features from ResNet models on CIFAR-10. We compare the protection against this attack for real-valued DNNs, real-valued DNNs with an additional layer, complex-valued DNNs using the complex-valued features $x$ and complex-valued DNNs using the approximated real-valued features $a^*$ estimated by the angle prediction model. We see that complex-valued networks are much more difficult to reconstruct inputs from, having error rates 2-3 times higher than real-valued networks. These results are in line with what was reported by the original authors.

Further experiments are performed with LeNet and ResNet-56-$\alpha$ on the CIFAR-10 and CIFAR-100 datasets and the results can be seen in Table 5. Again we see that complex-valued networks have higher error rates than real-valued

networks which follows what was originally reported. Noisy DNNs error rates fall between the original and complex DNNs which is the same as reported. However, the LeNet model on CIFAR-100 did not observe the same increase in MAE between the noisy and complex-valued networks.

| | Dataset | MAE (radian) |
|---|---|---|
| ResNet-20-$\alpha$ | CIFAR-10 | 0.0842 |
| ResNet-20-$\beta$ | CIFAR-10 | 0.0884 |
| ResNet-32-$\alpha$ | CIFAR-10 | 0.0875 |
| ResNet-32-$\beta$ | CIFAR-10 | 0.0874 |
| ResNet-44-$\alpha$ | CIFAR-10 | 0.0817 |
| ResNet-44-$\beta$ | CIFAR-10 | 0.0895 |
| ResNet-56-$\alpha$ | CIFAR-10 | 0.0839 |
| ResNet-56-$\beta$ | CIFAR-10 | 0.0846 |
| LeNet | CIFAR-10 | 0.2371 |
| LeNet | CIFAR-100 | 0.3178 |
| ResNet-56-$\alpha$ | CIFAR-100 | 0.0812 |

Table 3: Reproduction of average rotation angle error of complex-valued encoder features.

| | | Reconstruction Error (MAE) | | | |
|---|---|---|---|---|---|
| | Dataset | Original DNN | DNN with additional layers | Complex-Valued $\text{dec}(a^*)$ | Complex-Valued $\text{dec}(x)$ |
| ResNet-20-$\alpha$ | CIFAR-10 | 0.0750 | 0.1003 | 0.1904 | 0.2382 |
| ResNet-20-$\beta$ | CIFAR-10 | 0.0807 | 0.1009 | 0.2268 | 0.2865 |
| ResNet-32-$\alpha$ | CIFAR-10 | 0.0927 | 0.1872 | 0.1931 | 0.2544 |
| ResNet-32-$\beta$ | CIFAR-10 | 0.0748 | 0.2201 | 0.2050 | 0.2678 |
| ResNet-44-$\alpha$ | CIFAR-10 | 0.0735 | 0.2132 | 0.2219 | 0.2880 |
| ResNet-44-$\beta$ | CIFAR-10 | 0.0642 | 0.2192 | 0.2261 | 0.2994 |
| ResNet-56-$\alpha$ | CIFAR-10 | 0.0581 | 0.0866 | 0.2092 | 0.2832 |
| ResNet-56-$\beta$ | CIFAR-10 | 0.0582 | 0.0935 | 0.2459 | 0.3413 |

Table 4: Reproduction of inversion attacks against several ResNet variants on the CIFAR-10 dataset. For the attack $\text{dec}(a^*)$, we first estimate the real-valued features $a^*$ from the complex-valued features $x$.

| | | Reconstruction Error (MAE) | | | | | | |
|---|---|---|---|---|---|---|---|---|
| | Dataset | Original DNN | DNN with additional layers | Noisy DNN $\gamma = 0.2$ | Noisy DNN $\gamma = 0.5$ | Noisy DNN $\gamma = 1.0$ | Complex $\text{dec}(a^*)$ | Complex $\text{dec}(x)$ |
| LeNet | CIFAR-10 | 0.2070 | 0.2465 | 0.2108 | 0.2421 | 0.3637 | 0.4285 | 0.4423 |
| LeNet | CIFAR-100 | 0.1698 | 0.2374 | 0.2000 | 0.2042 | 0.2527 | 0.2836 | 0.2497 |
| ResNet-56-$\alpha$ | CIFAR-100 | 0.0830 | 0.0955 | 0.1005 | 0.1280 | 0.1489 | 0.1517 | 0.2135 |

Table 5: Reproduction of inversion attacks against LeNet and ResNet-56-$\alpha$ on the CIFAR-10 and CIFAR-100 datasets. For the attack $\text{dec}(a^*)$, we first estimate the real-valued features $a^*$ from the complex-valued features $x$.

### 3.3 Protection Against Inference Attacks

The property inference attacks are performed on a ResNet-56 prototype network that is trained on the 20 superclasses of CIFAR-100. The hidden properties that the attacker tries to infer are the standard 100 classes of CIFAR-100. Results of inference attacks 1, 2 and 3 can be seen in Figure 1. Inference attack 1 exhibits a $\sim 30\%$ error increase when using a complex-valued network compared to real-valued models. Inference attacks 2 and 3 similarly show better protection from complex-valued networks with $\sim 7\%$ error increase in both attacks. While our experiments show improved protection from complex-valued networks, it is much less significant than the $85\% +$ error the original authors reported on all three attacks.

## 4 Discussion

We will now discuss whether our implementation supports the main claims that Xiang et al. reported in their work.

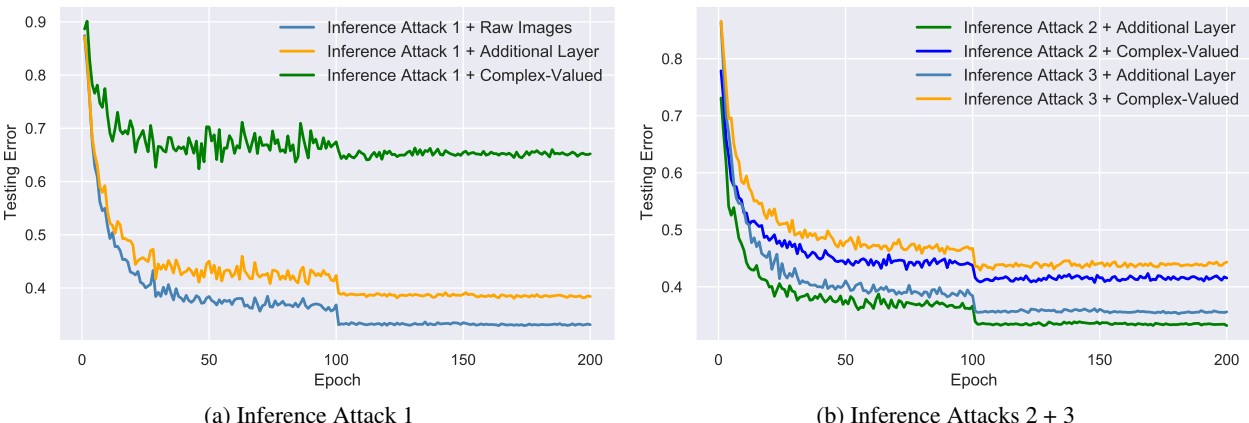

(a) Inference Attack 1  (b) Inference Attacks 2 + 3

Figure 1: Reproduction of inference attacks. Plots show the classification error rates of a ResNet-56 attacker model on the CIFAR-100 dataset. The prototype models are a complex-valued ResNet-56-$\alpha$ and a ResNet-56 with the additional layer trained on the superclasses of CIFAR-100.

**Complex-valued networks maintain similar performance as real-valued networks**  The authors report that complex-valued networks achieve similar or better classification performance across all the architectures and datasets they tested. We found this not to hold, with complex-valued networks performing consistently worse than their real-valued counterparts in our experiments. Deeper architectures such as ResNet-56 exhibit smaller performance losses compared to shallow architectures like LeNet which suggests that the conversion between real-valued and complex-valued features impacts the model's ability to learn and that a larger processing module helps to mitigate this. Since no training details are provided in the original work, it is difficult to identify if the choice of optimizer or hyperparameters led to these discrepancies. However, since we used reasonable training settings in all of our experiments, we are confident in fairness of the comparisons. We did not reimplement the experiments on higher resolution datasets such as VGG-16 [14] on CUB-200 [16] or AlexNet [8] on CelebA [10] which would help understand if our observations hold outside of low resolution datasets.

**Complex-valued networks are less susceptible to attacks**  We found that these claims do hold for complex-valued networks in both inversion and inference attacks, however not all results are as strong as what the authors originally reported. The most contradicting result is that our angle prediction model is capable of estimating the phase of complex-valued features $x$ within 0.1 radians on all ResNet variants. This is much lower than what was originally reported and indicates that $x$ does contain discriminative information about its rotation. This discrepancy may have stemmed from issues in the adversarial training or the training procedure for the attacker. The value of $k$ is not reported for the adversarial loss function, so we used $k = 5$ in all experiments. A much larger $k$ might result in less distinguishable encoder features, however, this would significantly increase computational time. It is also unclear whether the angle prediction model was trained separate or end-to-end with the inversion model. When initially testing an end-to-end approach, training was unstable and the angle prediction model only produced random results, even worse than what the authors reported. Training the angle prediction model separately performed significantly better and we report these results since it is a more realistic approach from an attacker.

Our results for the feature inversion attacks line up closely to what was originally reported in terms of MAE. Complex-valued networks have significantly worse reconstruction errors compared to real-valued networks, indicating that they are less vulnerable to inversion attacks. Since our angle prediction performs well, we find that using the estimated real-valued features is more effective than reconstructing from complex-valued features directly which contradicts what the authors report. We also observe a correlation between the classification performance of an architecture and the reconstruction error of the inversion attack. Networks that achieve better classification performance are shown to have lower reconstruction errors which can be seen when comparing LeNet and ResNet models on CIFAR-10. This suggests that architectures that produce better encoder features are more susceptible to inversion attacks. Inspecting the sample outputs for the LeNet variants in Figure 2, we can see that the reconstructions from complex-valued networks have less details and a slight tint from the original input, however, it is still evident that the reconstructions are simply distorted versions of the input image. In many situations, this level of obfuscation may not be sufficient for private information to be adequately protected. On the other hand, the visualizations presented by the authors are very strong with the reconstructions having no resemblance to the original image and containing many artifacts. The similar reconstruction errors and varying qualitative results may be caused by differences in how we measured error rates.

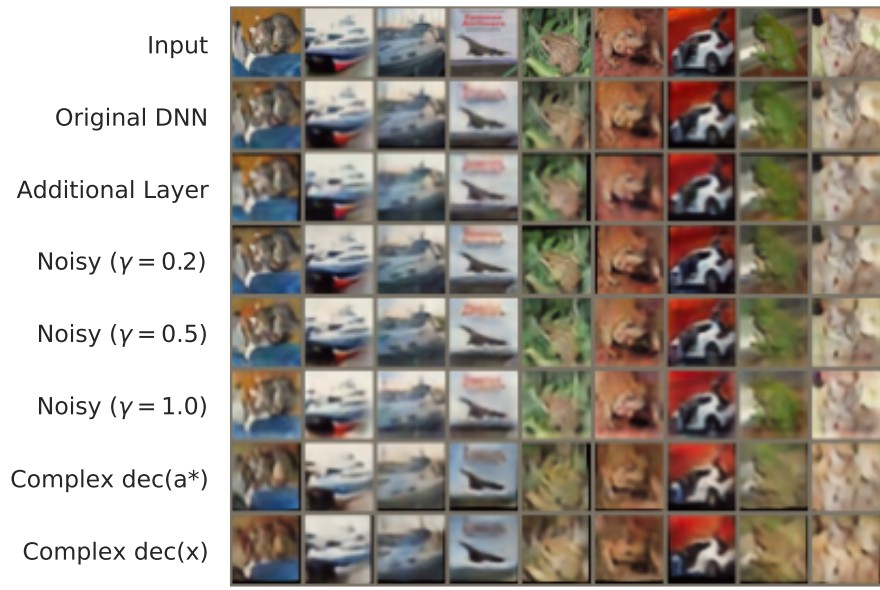

Figure 2: Visualizations of reconstructed CIFAR-10 test images from inversion attacks on the complex-valued LeNet and its real-valued baselines.

Inference attacks are similar to the inversion attacks, where we found that attacks performed worse on complex-valued networks compared to real-valued networks but not as effective as originally reported. Since the angle prediction and inversion models are utilized in inference attacks 2 and 3, it is expected that these attacks are more effective given that the inversion attacks in our implementation perform better than what is reported. Therefore, we cannot confirm the original claim that complex-valued DNNs prevent attackers from inferring any hidden properties, but we agree that they do hinder attackers ability to do so when compared with real-valued networks.

### 4.1 What was easy

It was fairly easy to follow the authors descriptions on how to split a real valued neural network into encoder, processing unit and decoder. Furthermore, most of the revised layers in the processing unit were also well described.

### 4.2 What was difficult

As no publicly available implementation was available, we were required to fully implement the complex-valued network, its training framework and all of the attacks which required a significant amount of time to complete. The original paper does not report any training details or model hyperparameters which made it impossible to reproduce the experiments exactly as reported. Other important details like exact architectures for the discriminator and angle prediction model and the training procedure for inversion attacks made the implementation difficult and require more trial and error than what we expected to encounter.

### 4.3 Communication with original authors

Due to lack of time, we did not have any communication with the original authors.

## 5 Conclusion

In this work we implemented the complex-valued DNN proposed by Xiang et al. Our reproduced experiments found that the complex-valued networks have consistently worse classification performance than their real-valued counterparts which contradicts what the authors report. We observed that the networks are less susceptible to feature inversion and property inference attacks however not as effective as originally described. The approach still provides some privacy protection, however, it may not be adequate in situations with very sensitive data.

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
