# OpenReview forum: "Interpretable Complex-Valued Neural Networks for Privacy Protection - ML Reproducibility Challenge 2020"
_ML_Reproducibility_Challenge/2020 — Reject_

### Official Review · AnonReviewer3 · 2021-03-01
**Review: Strong accept**

**Rating:** 9
**Confidence:** 5

**Review:**

1. The authors have clearly identified the following claims in the paper for reproducibility:
   1. Superior privacy preserving properties of the complex-values networks compared to the standard networks.
   2. Similar classification performance as real-valued networks on both datasets.
2. The authors of this report have implemented the code by themselves as the original implementation wasn't open-sourced. Despite the lack of information, the authors have also done hyper parameter sweep. The authors have clearly provided the implementation details in their report.
3. Due to lack of time they couldn't communicate with the original authors and have mentioned it in the report.
4. In their discussion and results section, the authors have clearly identified the parts of the paper that were easy to implement and those that weren't. Through extensive experiments, the authors have shown that while the privacy preserving claims of the complex networks hold, their classification performance is impacted compared to the baseline model. This is in contrast to the claim reported in the paper. However the authors have also considered if the difference in the accuracy might boil down to implementation of underlying architecture. Overall the authors have suggested that to further support the the main claim of privacy preservation, the paper can be augmented by adding pseudo-code and figures.

**Familiar With The Original Paper:**

I have read the original paper

**Reproducibility Summary:**

Report has summary

---

### Official Review · AnonReviewer2 · 2021-03-01
**Reproducing  Interpretable Complex-Valued Neural Networks for Privacy Protection**

**Rating:** 6
**Confidence:** 3

**Review:**

The reproducibility of this paper involved coding the experiments since no code was available from the original authors. Considerable effort has been taken to recreate the experiments as closely as possible but no effort was taken to contact the original authors for hyper-parameters to test the reported metrics. The reproducibility report, however, outlines the main claims made by the original authors and test these claims in their experiments. They have contrary results to the original claim that complex valued networks perform similar or better than real-valued networks. However, this is difficult to compare without training the network on the original parameters with the right optimiser. Moreover, the models have only been trained once which is not adequate to obtain optimal model performance and hence not adequate effort to report believable results.
The reproducibility report is able to verify that complex valued networks are less susceptible to inversion attacks but with weaker results. This could again be attributed to the difference in model parameters and optimisation technique used and hence an expectation of identical results is not reasonable.

**Familiar With The Original Paper:**

I have read the original paper

**Reproducibility Summary:**

Report has summary

---

### Official Review · AnonReviewer1 · 2021-03-09
**Good reproducibility report, but nothing significant**

**Rating:** 7
**Confidence:** 4

**Review:**

This report does reproducibility study of Xiang et al. ICLR 2020 Interpretable Complex-Valued Neural Networks for Privacy Protection.

Reproducibility Summary: present with adequate and sufficient summary of the performed reproducibility study.

Scope of reproducibility: is clearly stated and the report follows it

Code: the authors of the report implement the code from scratch as the original authors didn't make their code publicly available. The link to the repository is provided. All the results from the report seem to be able to be reproduced by following the easy to follow jupyter notebook. Although the code looks clean and readable from a quick glance, the documentation for the code is very limited or absent.

Communication with original authors: the authors of the report claim that they didn't communicate with the original authors due to lack of time. This claim without any further clarifications looks a bit odd as communication with the original authors may resolve some issues which the authors of the report mentioned as required lots of time, such as some missing implementation details in the original paper.

Hyperparameter Search: no hyperparameter search has been performed with discussion that this was due to limit of time. Considering the amount of experiments performed and that the implementation was done from scratch, the claim looks reasonable.

Ablation Study: no ablation study is performed beyond the original paper.

Discussion on results: The report explicitly discusses which parts were easy and difficult to reproduce and which claims from the original paper were confirmed in their experiments

Recommendations for reproducibility: though there are no explicit recommendations the authors emphasise which implementation details were missing in the original paper that caused the most problems in the reproducibility study

Results beyond the paper: No results beyond the paper, moreover some of the more computationally heavy results from the original paper were not reproduced

Overall organization and clarity: The report is very well written and easy to follow

**Familiar With The Original Paper:**

I have not read the original paper

**Reproducibility Summary:**

Report has summary

---

### Decision · Program_Chairs · 2021-03-31

**Decision:**

Reject

**Comment:**

The reproduction effort is considerable, but the actual report is fairly limited, and it is hard to interpret the contribution of the results given the way they are presented.